# Electroporation in Translational Medicine: From Veterinary Experience to Human Oncology

**DOI:** 10.3390/cancers16051067

**Published:** 2024-03-06

**Authors:** Enrico P. Spugnini, Maria Condello, Stefania Crispi, Alfonso Baldi

**Affiliations:** 1Biopulse Srl, 00144 Rome, Italy; alfonsobaldi@tiscali.it; 2Istituto Superiore di Sanità, 00161 Rome, Italy; maria.condello@iss.it; 3Institute of Biosciences and BioResources-UOS Naples CNR, Via P. Castellino 111, 80131 Naples, Italy; stefania.crispi@ibbr.cnr.it; 4Department of Environmental, Biological and Pharmaceutical Sciences and Technologies, Campania University “Luigi Vanvitelli”, 81100 Caserta, Italy

**Keywords:** electrochemotherapy, electroimmunotherapy, irreversible electroporation, cancer vaccine

## Abstract

**Simple Summary:**

The application of electric pulses to promote the delivery of anticancer molecules in in vitro and in vivo models is a widely accepted technique known as electroporation (EP). This technique has been diffusely adopted to increase the effectiveness of different antitumor therapies both in pets and in humans. Moreover, this approach has been instrumental to devise and develop novel charging systems to generate nanovesicles embedded drugs and vaccines. This article summarizes the state of art of different EP applications, such as electrochemotherapy (ECT), irreversible electroporation, and EP-based cancer vaccines, in veterinary and human oncology, describing the most recent progresses and the most striking obtained clinical results.

**Abstract:**

Electroporation (EP) is a broadly accepted procedure that, through the application of electric pulses with appropriate amplitudes and waveforms, promotes the delivery of anticancer molecules in various oncology therapies. EP considerably boosts the absorptivity of targeted cells to anticancer molecules of different natures, thus upgrading their effectiveness. Its use in veterinary oncology has been widely explored, and some applications, such as electrochemotherapy (ECT), are currently approved as first-line treatments for several neoplastic conditions. Other applications include irreversible electroporation and EP-based cancer vaccines. In human oncology, EP is still mostly restricted to therapies for cutaneous tumors and the palliation of cutaneous and visceral metastases of malignant tumors. Fields where veterinary experience could help smooth the clinical transition to humans include intraoperative EP, interventional medicine and cancer vaccines. This article recapitulates the state of the art of EP in veterinary and human oncology, recounting the most relevant results to date.

## 1. Introduction

Electroporation is a technique used to increase the transmembrane flow of molecules of interest (either for basic science or for therapeutic purposes) in plant or mammalian cells through exposure to electric fields having fixed parameters [1]. The alteration of membrane permeability can lead to two different outcomes: (1) cessation of the condition of enhanced permeability and restoration of the initial condition and (2) inability of the cell to revert to the previous ionic balance and activation of the apoptotic or necrotic pathways, ultimately resulting in osmotic death. The first condition is called reversible electroporation, while the second is irreversible electroporation [2]. Electric pulse (EP)-based treatments exploit the properties of electric fields to therapeutically hit cancer cells and broaden the options available to clinicians. EP-based therapies include irreversible electroporation (IRE), gene electrotransfer (GET), electrochemotherapy (ECT), calcium electroporation (Ca–EP) and therapeutic vesicles loading (i.e., liposomes, exosomes, etc.) (EP–TVL). Unfortunately, these approaches are still uncoordinated and these therapies are still confined to a limited number of clinical applications. This review aims to summarize the current state of EP-based technologies, highlighting mechanisms of action, medical indications, technology progresses and avenues of investigation. Membrane permeabilization includes five moments: induction, expansion, stabilization, resealing and memory effect [1]. Shape, number and duration of the electric pulses are among the most crucial factors influencing membrane electroporation [1]. In detail, EP initially induces a phase of pore creation, followed by pore expansion [3]. The initial membrane disruptions are called “transient electropores”. After a given time, they shrink and temporarily stabilize, becoming “long-lasting electropores” [3]. Transient electropores are of paramount importance to the cross-membrane flow of large molecules (larger than several kiloDaltons). Smaller molecules are carried on during both periods, with a greater quantity moved during the long-lasting period. As a result, the correct timing administration of the different molecules is crucial for a proficient transfer to cellular targets. Regarding the influence of EP shape, the most effective waveforms for the different applications (laboratory versus clinical) are electric square and biphasic pulses, while exponentially decaying pulses are less proficient [1].

## 2. Different Electroporation Techniques

IRE is a technology that uses a DC current up to 3 kV to induce cell death. In this setting, EPs are applied in up to 60–100 high-voltage (1.5–3 kV) bursts of 80–100 µs to irreversibly permeate the cell membrane, leading to altered ion fluxes that ultimately, through a calcium overloading mechanism, result in the activation of the reactive oxidative species and osmotic death. The cell death caused by IRE occurs without appreciable thermal warming or heat-induced tissue damage [4]. Notably, this technology does not hinder the surrounding nerve fibers [5] and tissue scaffolding, while at the same time abating the “heat sink effect” [6], a phenomenon that thwarts the efficiency of other therapies when the target lesion is near (<1 cm) a large vessel (≥3 mm in diameter); the blood flow can indeed promote a cooling effect, thus decreasing the tangible ablation volume.

GET exploits the induction of transient electropores to drive genetic particles, such as plasmids, within the targeted cell, leading to cell death or the activation of the immune system [7].

ECT is a treatment that associates the local or systemic injection of chemotherapy drugs with short, high-voltage electric pulses of appropriate waveform and pre-determined amperage [3]. The application of electric pulses to the tumor tissues permeabilizes the tumor cell, overcoming membrane resistance to the passage of lipophobic drugs such as bleomycin and cisplatin. This results in improved effectiveness that permits a decrease in the doses of chemotherapy and the confining of side effects while preserving clinical efficacy [8]. There are variations in drug absorption that are influenced by the nature of the different histotypes, as evidenced by meta-analysis studies [9] that suggest that this is especially dependent on their microvascularization [10]. Recently, thanks to the development of a combined injection of bleomycin with indocyanine green in veterinary oncology cases, as well as a novel pharmacokinetic method to evaluate the distribution of bleomycin in plasma, serum and tissue, dose adjustments have been possible, especially in elderly humans [11,12]. Several studies have proved the clinical validity of this therapy in terms of enhanced tumor control, with overall limited toxicity in both human and veterinary medicine [1,8].

Ca–EP is an investigational anti-cancer therapy that drives high calcium concentrations within tumor cells through electroporation [13]. Again, it exploits the property of high-voltage pulses to prompt transient permeabilization of the plasma membrane by enhancing the flow of ions into the cytosol. Calcium is a strictly controlled, universal cell mediator pivotal in many cellular reactions, including, among others, the activation of the reactive oxidative species, ultimately leading to cell apoptotic death. Through the increase in its concentration, EP can induce severe cellular stress that results in ATP depletion, leading to cancer cell death [14,15].

EP–TVL is a novel application of electroporation, with a potential secondary positive impact on clinics. EP allows the loading of nanovesicles with different agents having anticancer properties (chemotherapy agents, toxins, antibodies, plant-derived molecules), permitting a smoother transmembrane transition with potentiation of anti-tumor effectiveness [16]. All these EP applications are summarized in Figure 1.

## 3. Translational Use of EP in Veterinary Oncology and Humans

### 3.1. Irreversible Electroporation (IRE)

IRE has been extensively studied over the past years as a promising therapy for inoperable or advanced metastatic cancers affecting different organs, including the prostate [17,18], pancreas [19,20], liver [21,22] and breast [23]. Notably, preclinical studies have elucidated critical operational features, such as the need for neuromuscular blocking to decrease the muscle contractions caused by electrical stimulation and ECG synchronization to avert the induction of cardiac arrhythmias.

The investigation of IRE in pets has been somewhat limited but has provided insights into specific districts, specifically the brain, where the feasibility of IRE in this sensitive organ has been proven in two canine studies [24,25]. In the first investigation, six dogs with glioma were treated with IRE. An objective response was reported in four out of six veterinary patients, obtaining a survival time ranging from 1 to >940 days. The second study proved that IRE can be applied to brain neoplasms incorporating areas closest to critical vasculature and can supply clinically significant volumes of tumor removal. Interestingly, the foci of mineralization may interfere with the obtainment of complete tumor ablation. It is foreseeable that these studies will speed up the use of IRE for the treatment of telencephalic gliomas in humans. Another histotype that might benefit from canine investigation is urethral transitional cell carcinoma [26]. This neoplasm can cause severe renal impairment due to obstruction of urine flow, with high morbidity and mortality among the affected individuals. A recent investigation in dogs evidenced the feasibility of transurethral IRE using a dedicated balloon catheter, generating data easily transferable to humans. Finally, transcutaneous IRE has been used to attack unresectable liver tumors in a canine model [27]. In this study, IRE was safely and effectively delivered percutaneously, resulting in an anticipated ablation volume with secondary lymphocytic tumor infiltration. In conclusion, it is possible to claim that IRE involves specific expertise in determining the correct needle electrodes and the most appropriate electric parameters. Indeed, the use of incorrect parameters can greatly reduce the treatment efficacy. Therefore, it is compulsory to define a standard operative procedure and to approach this technology within a multidisciplinary team.

### 3.2. Gene Electrotransfer (GET)

One study demonstrated that genes can be transferred in mouse muscle and expressed for prolonged periods of time in vivo [28]. This study encouraged gene transfer procedures on in vivo models by viral and non-viral vectors for vaccination and gene therapy [29,30]. The use of viral vectors has the important benefit of warranting high transfection efficiency and reliability [31]. However, there were some drawbacks, including excessive costs, cytotoxicity, the likelihood of triggering mutagenesis and a restriction on the sizes of the transgene to be inserted [32]. In contrast, non-viral vectors are considerably less expensive, allow the insertion of larger transgenes, exhibit lower cytotoxicity and immunogenic responses, and reduce the risk of insertional mutagenesis [30]. Nonetheless, plasmid-based DNA transfection by direct injection into skeletal muscle has very low proficiency [33,34], and this tightens its experimental and therapeutic use for in vivo applications.

The low transfection efficiency of plasmid DNA through direct injection may be linked to the structure of plasmid constructs, injection volume, vehicle type and injection speed [35].

Different experimental strategies have been adopted to ameliorate transfection efficacy, including the adoption of solutions able to inhibit nuclease-mediated digestion of DNA, various inoculation parameters and the application of an electric field on the injection site [36,37].

In vivo studies have supported the fact that the application of an electric field greatly improves transgene expression in skeletal muscle [38]. Nevertheless, a few electroporation protocols have been recommended, but many variables have been observed that modify transfection efficiency [32,39]. The most important variables to be considered include the voltage applied, the length of electric pulses, the number of electric pulses, the pre-treatment of the muscle, and the DNA vehicle. Finally, it is crucial to evaluate the injury caused by electroporation on the treated muscles, which can reduce the efficiency and repeatability of transfection [32]. GET has been adopted in veterinary oncology with two approaches: (1) to directly attack neoplasms, such as mast cell tumors or melanoma, by combining electroporation with the local injection of plasmids encoding for interleukin 12 [40,41,42,43,44,45]; and (2) to delay or prevent tumor recurrence or metastases after chemotherapy or surgical excision through an electroporation-mediated vaccination [46,47,48,49]. Both approaches have proven successful in terms of local control and overall survival, providing evidence that, if corroborated by further results, might speed up its transition to humans.

### 3.3. Electrochemotherapy (ECT)

#### 3.3.1. Clinical Electrochemotherapy Protocols in Veterinary Oncology

This is the applicative field of electroporation that has focused the interest of basic scientists and clinicians in veterinary medicine over the past 20 years. As a result, veterinary investigations have moved independently and parallel those of humans. 

Three major aspects underline the independence of veterinary ECT from human ECT [8]:The only standardized and approved protocols for human cancer treatment are those for the palliative care of cutaneous cancer metastases or for the treatment of primary skin tumors. There are many investigations of the treatment of visceral cancer through endoscopy or laparoscopy, but they are not yet approved standard approaches. On the other hand, in veterinary oncology, ECT has been adopted as a first-line treatment of solid tumors and for the treatment of selected visceral neoplasms, to the point that such therapies are reimbursed by veterinary insurance schemes.Although human ECT is habitually performed under local anesthesia, veterinary ECT is performed with the patients under heavy sedation or general anesthesia.Veterinary ECT can be palliative, adjuvant or neoadjuvant and can be administered concurrently with surgery (intraoperative ECT). Additionally, ECT guided by ultrasound is gaining importance in veterinary oncology for treating deep tumors that are not easily reachable by surgery or that are at an advanced stage.

#### 3.3.2. Clinical Outcome in Solid Tumors

##### Feline and Canine Soft Tissue Sarcoma (STS)

Currently, ECT is adopted for the adjuvant therapy of incompletely excised STS in cats and dogs. Cats are an important model for humans since STS in this species is often extremely aggressive, rapidly growing and difficult to control [50,51]. ECT has shown great effectiveness in preventing or delaying STS recurrence in both species after incomplete surgical excision in both intraoperative and postoperative modalities [52,53,54,55]. ECT has been instrumental in reintroducing drugs with narrow therapeutic indices, such as cisplatin in cats, into the veterinary protocols, thus opening a path for the possible reintroduction of similar agents into both veterinary and human oncology [8]. Finally, veterinary oncologists have successfully investigated the possibility of using combination chemotherapy in pets, thus reinforcing the efficacy of the currently adopted protocols [56,57].

##### Epithelial Tumors

Head and neck carcinoma is a common neoplasm in white-skinned humans and white-haired pets [58,59]. White cats are extremely vulnerable to ultraviolet carcinogenesis and tend to develop squamous cell carcinoma in the ear, nasal planum, eyelid and head skin. Cats (especially stray cats) are frequently referred with advanced diseases that pose a significant challenge to veterinarians. In recent years, ECT has proven to be an effective alternative to standard therapeutic strategies (surgery and/or radiation therapy), showing limited side effects, a high degree of success and financial sustainability for owners [60,61,62,63,64,65]. An interesting success rate has been described in a case series of nontonsillar canine oral squamous cell carcinoma [66]. Cutaneous squamous cell carcinoma has also shown a high degree of responsiveness [67]. Other epithelial tumors approached fruitfully in veterinary oncology include carcinoma of the perianal and anal sac glands in dogs. These neoplasms, due to their anatomic location adjacent to nerves and a muscular sphincter, make their control a clinical challenge. ECT has been able to control these neoplasms either by direct application or through ultrasonographic guidance, resulting in prolonged survival for the dogs [68,69,70]. Figure 2 and Figure 3 provide two examples of successfully treated epitehelial tumors in cats.

##### Melanoma

Melanoma is frequently reported in dogs (with oral, ocular and digital localization) and in gray horses. It was the first round cell tumor to be directly attacked with ECT. The results evidenced the feasibility of these techniques in anatomically and clinically challenging conditions, obtaining a high degree of local control and, in dogs, a reduction of distant metastases due to stimulation of the immune system by uncovering the tumor antigens through the administration of plasmids encoding for canine interleukin 12 [44,45,71,72,73].

**Figure 3 cancers-16-01067-f003:**
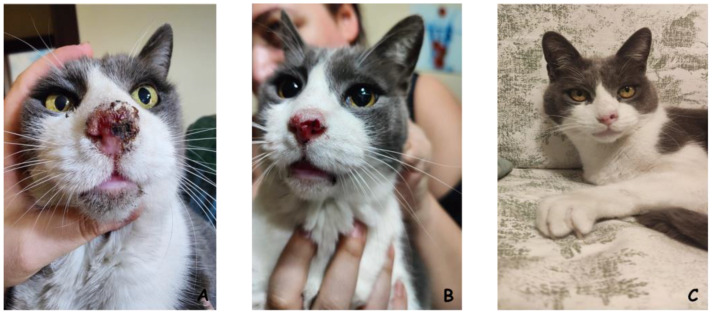
A 12-year-old male neutered domestic short-haired cat with squamous cell carcinoma of the nasal planum. (**A**) The cat at presentation, showing an advanced tumor lesion with ulceration and remodeling of the affected area. (**B**) The cat after one session of electrochemotherapy; the tumor has regressed, leaving a much smaller neoplastic erosion. (**C**) The cat at the completion of the electrochemotherapy course (three sessions, two weeks apart from each other); there is no longer an appreciable tumor, and the nasal planum has gained its previous appearance. There is mild tissue loss due to tumor destruction by electrochemotherapy. (Spugnini EP, personal observation).

### 3.4. ECT Clinical Trials in Humans

Many studies on electroporation are ongoing, with 306 studies registered on the website clinicaltrial.gov (https://clinicaltrials.gov/search?term=electroporation&city=, accessed on 2 December 2023). Many clinical studies have demonstrated the efficiency and safety of ECT on different tumors (melanoma, colorectal, gynecological, pancreatic, liver, head and neck cancer) (Table 1).

The NCT00006035 clinical trial demonstrated the effectiveness of ECT and bleomycin in stage III or IV melanoma patients, comparing the recovery time of patients treated with ECT plus bleomycin with those treated with the chemotherapy alone (https://clinicaltrials.gov/study/NCT00006035, accessed on 2 December 2023).

Another clinical study recruited 30 patients with 654 cutaneous and subcutaneous metastatic melanomas. After 24 months of treatment with intravenous bleomycin ECT, the local tumor control rate was 72%; histological analysis showed partial regression of melanomas, suggesting a promising application of ECT in metastatic melanoma treatment [74].

A prospective cohort study performed on 151 patients with metastatic melanomas demonstrated that treatment with ECT and bleomycin induced a complete response in 58% of lesions [75].

A phase II multicenter, open-label, non-randomized study (NCT03448666) on 53 patients with unresectable melanomas and superficial and visceral metastases aimed to verify that the combination of Pembrolizumab with ECT is safe and able to improve local and systemic efficacy. The first results are expected by the end of 2023 (https://clinicaltrials.gov/ct2/show/NCT03448666, accessed on 2 December 2023). 

The first phase I study on seven patients with colorectal tumors used endoscopic electroporation with bleomycin. Magnetic resonance imaging and computed tomography showed the immediate cessation of bleeding and the reduction of intraluminal masses after a short treatment time [76]. A similar endoscopic device for electroporation has been used in a phase I study to treat patients with esophageal cancer [77]. In addition, the treatment was well tolerated in this model, and tumor response was visualized in five of six patients.

A new randomized controlled trial (NCT04816045) on patients with histologically verified rectal and sigmoid colon cancer is in progress. Clinical examination results, blood samples, biopsies and questionnaires will be collected to evaluate safety and tumor and immunological responses to ECT. The results will be published at the end of 2023 and will be based on an examination of blood samples and biopsies to verify the safety of ECT and the improvement of therapeutic response (https://clinicaltrials.gov/study/NCT04816045, accessed on 2 December 2023).

ECT could be used for gynecological tumor treatment. In 2021, a phase II clinical trial (NCT04760327) opened the recruitment of patients to evaluate the effectiveness and safety of combined treatment electroporation with bleomycin or cisplatin, respectively, according to RECIST1.1 and CTCAEv5.0 criteria (https://clinicaltrials.gov/study/NCT04760327, accessed on 2 December 2023).

Clinical trials on pancreatic and liver tumors using a Cliniporator device were performed (NCT04281290, NTC02291133). The first, the PanECT study, included 20 patients in a phase I clinical study and an additional 20 patients in a phase II clinical study (https://clinicaltrials.gov/study/NCT04281290, accessed on 2 December 2023). The second included 10 patients in a phase I clinical study and an additional 15 patients in a phase II clinical study (https://clinicaltrials.gov/study/NCT02291133, accessed on 2 December 2023).

Overall, the main clinical studies, including those on solid tumors of various histotypes, confirmed that the cytotoxicity of bleomycin improved after electroporation compared to the drug alone. Moreover, this technique was shown to be safe. The treatment conduct was standardized in the framework of the European Standard Operating Procedure on ECT (ESOPE), released in 2006 [78] and recently updated [79]. All these studies are reassuring and establish ECT as an alternative methodology to classical chemotherapy to overcome tumor drug resistance.

### 3.5. Calcium–EP

Calcium–EP is another treatment modality based on electroporation that induces a high calcium concentration within the cells. The high intracellular calcium concentration induced a decrease in ATP levels, mitochondrial dysfunction and cell death [14].

In human oncology, preliminary investigations have pointed out that intratumoral calcium is as effective as bleomycin for the treatment of cutaneous metastatic neoplasms and for the treatment of recurring head and neck cancer [80,81].

In veterinary oncology, Calcium–EP has been used as a first line of treatment in horses with cutaneous sarcoids, obtaining high levels of local control with limited side effects. These results have led to the belief that this strategy could easily become a first line of treatment in veterinary oncology and be more widely accepted in humans as well [82,83].

Our group is currently investigating the effectiveness of local calcium injections in enhancing the efficacy of bleomycin-based ECT.

### 3.6. EP–TVL

Electroporation-based therapeutic vesicles loading is not a therapy per se, since electroporation is instrumental in implementing the charging of vesicles with anticancer agents of various natures (e.g., chemotherapy, photodynamic agents, antibodies, mi-RNA, ricin, etc.) [84]. These products show promise as highly efficient therapeutic approaches to neoplasms [85]. Extracellular vesicles (EVs) are membrane-bound vesicles produced and subsequently released into the extracellular milieu by vegetal and animal cells. EVs work as nutrient carriers, acting as shuttles to transfer signal transductors of various natures to neighboring and distant cells. Among the EVs, three categories captured the attention of researchers: exosomes, liposomes and synthetic nanovesicles. EVs have several advantages over other therapeutic strategies because they allow the precise loading of molecules and prevent degradation. They also have improved drug delivery proficiency and the capacity to move easily through biological barriers. Moreover, membrane manipulation through the insertion of specific ligands can enhance the specificity of targeted delivery.

Exosomes have recently been identified as an alternative framework within the tumor niche and metastatic cascade [86,87]. Exosomes are composed of a lipid bilayer ferrying an organic cargo made up of miRNAs, mRNAs, proteins, lipids and other metabolites. Upon internalization by the target cells, they release their cargo into the cytoplasm. Tumor cell-derived exosomes showed tumor growth suppressor properties [88]. Furthermore, hepatic- and macrophage-derived EVs displayed antiviral activity [89].

EVs of mesenchymal origin have shown the capability to modify tissue healing and regeneration [90]. Although EVs have hopeful therapeutic potential, there are various problems associated with using EVs before transitioning from the laboratory to clinical use. Some of these challenges involve low loading efficiency with anticancer molecules, low yield, and difficulties in isolation and purification methodologies. Methods to charge EVs take in incubation, electroporation, sonication, extrusion, hypotonic dialysis, freeze-thaw cycles, saponin, CaCl2 and lipofectamine reagent. Among the others, electroporation is the technique raising the interest of researchers and biotech companies due to its effectiveness, preservation of membrane integrity, cargo yielding and its potential to be industrially exploited. Nevertheless, there is no agreement on the best technology, electrical protocol or standard procedures [84].

Liposomes (self-assembled phospholipids to obtain artificial lipid bilayers used to generate artificial vesicles) and other synthetic vesicles, different from EVs, are generated in vitro and are not naturally involved in any physiological process [91]. These new vesicles imitate the biophysical and biochemical features of EVs to bypass present limitations to the adoption of exosomes for therapeutic applications [92]. Again, several loading strategies have been proposed for these tools, and electroporation has been evaluated as being among the most promising techniques in terms of encapsulation efficiency [93].

## 4. Conclusions

Over the last 30 years, electroporation has shown a bounty of highly effective applications in medicine, ranging from laboratory engineering of EP charged particles to clinical applications in human and veterinary oncology [1,2,8]. There are few fields of medicine where the synergy between laboratory investigations and clinical translations has been more effective [94]. Recent articles and reviews on EP have clearly shown its expanding applications, rationalization of protocols and investigation for use in different cancer conditions. Electroporation utilizes electric fields that can be delivered using generators of novel ideation and by exploiting different materials to maximize the clinical outcome while at the same time minimizing side effects [94]. When dealing with cancer therapy, irreversible electroporation (IRE) is effective in killing cancer cells, while reversible electroporation (RE) can promote effective intracellular drug delivery. In particular, the combination of electroporation and drug treatment (ECT, electrochemotherapy) has proven to be effective, safe and well tolerated by the patient. The possibility of treating the tumor mass with a lower dose of drug reduces the toxic effects in healthy organs, with notable improvements in the patient’s quality of life and, moreover, has important repercussions on the national healthcare system for the reduction of drug costs. Additionally, the development of more sophisticated computers will help operators achieve greater precision, increasing its development in the field of interventional medicine, thus allowing safer tumor eradication from its host. Much still needs to be done, but the future of this technology appears bright and ripe with promise.

## Figures and Tables

**Figure 1 cancers-16-01067-f001:**
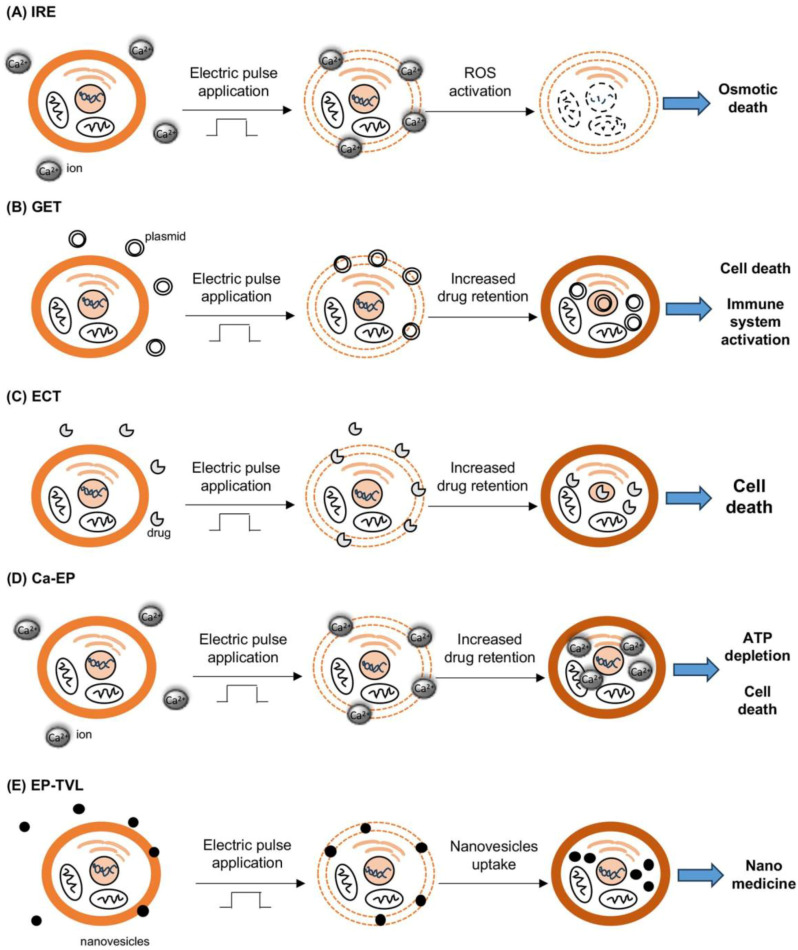
A schematic depiction of the applications of EP in oncology.

**Figure 2 cancers-16-01067-f002:**
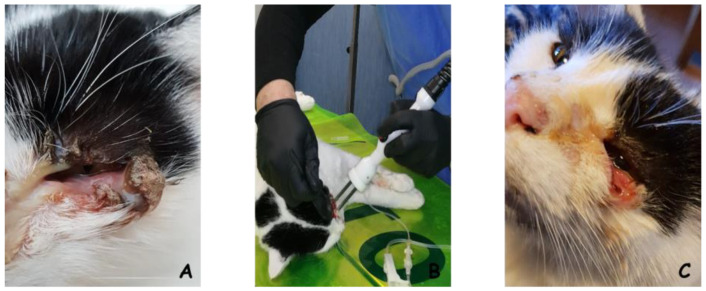
A 10-year-old female spayed domestic short-haired cat with palpebral squamous cell carcinoma. (**A**) Patient at presentation. (**B**) Patient receiving a session of electrochemotherapy under anesthesia. An eye shielding is placed on the eye globe to prevent corneal burns during the procedure. (**C**) The patient upon completion of the therapy. The tumor regressed, leaving a very thin layer of skin on the eyelid. (Spugnini EP, personal observation).

**Table 1 cancers-16-01067-t001:** Clinical trials of electrochemotherapy.

Study Title	Phase Study	Treatment Modality	Clinical Trials.gov Identifiers/References
Bleomycin with or without Electroporation Therapy in Patients with Stage III or Stage IV Melanoma	I	Electroporation plus bleomycin	NCT00006035
Electrochemotherapy: An Effective Local Treatment of Cutaneous and Subcutaneous Melanoma Metastases	I	Electroporation with Cliniporator plus bleomycin	Ricotti et al., 2014 [74]
Electrochemotherapy in the Treatment of Metastatic Malignant Melanoma: A Prospective Cohort Study by InspECT		Electroporation plus bleomycin	Kunte et al., 2017 [75]
ECT-Pembrolizumab in Patients with Unresectable Melanoma with Superficial or Superficial and Visceral Metastases	II	Electroporation with Cliniporator plus Pembrolizumab and bleomycin	NCT03448666
Electrochemotherapy for Colorectal Cancer Using Endoscopic Electroporation: A Phase 1 Clinical Study	I	Electroporation plus bleomycin	Hansen et al., 2020 [76]
Neoadjuvant Electrochemotherapy for Colorectal Cancer—a Randomized Controlled Trial	II	Electroporation plus bleomycin	NCT04816045
Electrochemotherapy of Gynecological Cancer (GynECT)	II	Electroporation plus bleomycin or cisplatin	NCT04760327
Electrochemotherapy of Posterior Resection Surface for Lowering Disease Recurrence Rate in Pancreatic Cancer (PanECT Study)	I/II	Electroporation with Cliniporator Vitae plus bleomycin	NCT04281290
Endoscopic-assisted Electrochemotherapy in addition to Neoadjuvant Treatment of Locally Advanced Rectal Cancer	II	Electroporation with EndoVE plus bleomycin	NCT03040180
Treatment of Primary Liver Tumors with Electrochemotherapy (ECT-HCC)	I/II	Electroporation with Cliniporator Vitae plus bleomycin	NCT02291133
Electrochemotherapy for Non-curable Gastric Cancer	I	Electroporation plus bleomycin	NCT0413907
Electrochemotherapy on Head and Neck Cancer	II	Electroporation with Cliniporator plus bleomycin	NCT02549742
Electrochemotherapy for Chest Wall Recurrence of Breast Cancer: Present Challenges and Future Prospects	II	Electroporation with Cliniporator plus bleomycin	NCT000744653

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
