# Peer review of "Electroporation in Translational Medicine: From Veterinary Experience to Human Oncology"

_cancers, 2024, doi:10.3390/cancers16051067_

Round 1

Reviewer 1 Report

Comments and Suggestions for Authors

The paper provides a general overview of electroporation (EP) in veterinary and human oncology.
However, there are some missing pieces, and paper requires revisions before the acceptance

- in figures, add information about the study, currently, it is not known if it is own study or not, it should be clarified

- in this review paper, there are numerous auto-citations, which is not acceptable. There are a lot of groups that are also working with ECT in veterinary, and the review paper requires a broadened view of this method and the involvement of various approaches.

- there is no data about the drug distribution during ECT procedures and how to control this. See research on combining bleomycin with indocyanine; despite this, indocyanine is widely used in veterinary oncology.

- describing GET, you should also mention about IL-2, IL-12 delivery and what type of plasmid are used,

- It would be beneficial to include any limitations or constraints in the current understanding or application of EP in veterinary and human oncology. This would provide a more balanced perspective on the state of the art.

Author Response

REVIEWER 1

The paper provides a general overview of electroporation (EP) in veterinary and human oncology.

However, there are some missing pieces, and paper requires revisions before the acceptance

- in figures, add information about the study, currently, it is not known if it is own study or not, it should be clarified

We provided the requested information, stating that they are unpublished personal observations

- in this review paper, there are numerous auto-citations, which is not acceptable. There are a lot of groups that are also working with ECT in veterinary, and the review paper requires a broadened view of this method and the involvement of various approaches.

We reported also the contribution of several other groups in the veterinary ECT section, however we implemented their number as requested.

- there is no data about the drug distribution during ECT procedures and how to control this. See research on combining bleomycin with indocyanine; despite this, indocyanine is widely used in veterinary oncology.

We thank the reviewer for the suggestion. We added these concept to the review

- describing GET, you should also mention about IL-2, IL-12 delivery and what type of plasmid are used,

We thank the reviewer for the suggestion. We implemented the GET section with the requested information

- It would be beneficial to include any limitations or constraints in the current understanding or application of EP in veterinary and human oncology. This would provide a more balanced perspective on the state of the art.

A section with the requested concepts have been added

Reviewer 2 Report

Comments and Suggestions for Authors

The reviewer finds the review article titled “Electroporation in translational medicine: from veterinary experience to human oncology” well written and highly informative. I find the contents on the state of the art of EP in veterinary oncology relevant as some of the lesions learned from companion animals are playing a crucial role in clinical trials and lead to clinical benefits. This is a much-needed update from the 2012 Book by Enrico P Spugnini titled “Electroporation in Laboratory and Clinical Investigations”.

However, given the wide range of EP techniques covered in this review, in addition to the overlapping contents found in this review with two other respectful reviews by the authors published on ECT, the reviewer would recommend the authors focus the reviewing on “EP on veterinary oncology” with the authors extensive background, other than attempting to cover the entire field of EP in oncology, with the following two reasons.

1. The lack of focus

The authors attempted to cover almost the entire field of EP-based therapies on oncology, including irreversible electroporation (IRE), gene electrotransfer (GET), electrochemotherapy (ECT), calcium electroporation (Ca-EP), and therapeutic vesicles loading (i.e. liposomes, exosomes etc) (EP-TVL), which may take an entire book.

However, most of the areas have their corresponding insightful reviews already, and the reviewers can hardly find anything that hasn’t already been covered by other authors. For example:

·         Covering different techniques of electroporation (section 2 of this review)

Geboers, B., Scheffer, H. J., Graybill, P. M., Ruarus, A. H., Nieuwenhuizen, S., Puijk, R. S., ... & Meijerink, M. R. (2020). High-voltage electrical pulses in oncology: irreversible electroporation, electrochemotherapy, gene electrotransfer, electrofusion, and electroimmunotherapy. Radiology, 295(2), 254-272.

·         IRE (section 3.1)

Aycock, K. N., & Davalos, R. V. (2019). Irreversible electroporation: background, theory, and review of recent developments in clinical oncology. Bioelectricity, 1(4), 214-234.

·         GET (section 3.2)

Heller, L. C., & Heller, R. (2021). Gene electrotransfer. Electroporation in Veterinary Oncology Practice: Electrochemotherapy and Gene Electrotransfer for Immunotherapy, 219-234.

·         Calcium-EP (section 3.5)

Frandsen, S. K., Vissing, M., & Gehl, J. (2020). A comprehensive review of calcium electroporation—A novel cancer treatment modality. Cancers, 12(2), 290.

Romeo, S., Frandsen, S. K., Gehl, J., & Zeni, O. (2019, June). Calcium electroporation: an overview of an innovative cancer treatment approach. In 2019 PhotonIcs & Electromagnetics Research Symposium-Spring (PIERS-Spring) (pp. 2979-2984). IEEE.

·         EP-TVL (section 3.6)

Lennaárd, A. J., Mamand, D. R., Wiklander, R. J., El Andaloussi, S., & Wiklander, O. P. (2021). Optimised electroporation for loading of extracellular vesicles with doxorubicin. Pharmaceutics, 14(1), 38.

However, the veterinary applications of IRE, GET and Calcium-EP have not been extensively reviewed yet. Therefore, the reviewer would recommend shifting the focus a bit.

2. Lack of original contribution

The following two reviews have covered the ECT from veterinary experience to human oncology exceedingly well. The reviewer has not able to identify too much new information on ECT in this reviewer that hasn’t been covered before, even though a few different examples were given.

Spugnini, E. P., Menditti, D., De Luca, A., & Baldi, A. (2023). Electrochemotherapy in translational medicine: from veterinary experience to human oncology. Critical Reviews™ in Eukaryotic Gene Expression, 33(1).

Spugnini, E. P., & Baldi, A. (2019). Electrochemotherapy in veterinary oncology: state-of-the-art and perspectives. Veterinary Clinics: Small Animal Practice, 49(5), 967-979.

Author Response

Reviewer 2

The reviewer finds the review article titled “Electroporation in translational medicine: from veterinary experience to human oncology” well written and highly informative. I find the contents on the state of the art of EP in veterinary oncology relevant as some of the lesions learned from companion animals are playing a crucial role in clinical trials and lead to clinical benefits. This is a much-needed update from the 2012 Book by Enrico P Spugnini titled “Electroporation in Laboratory and Clinical Investigations”.

However, given the wide range of EP techniques covered in this review, in addition to the overlapping contents found in this review with two other respectful reviews by the authors published on ECT, the reviewer would recommend the authors focus the reviewing on “EP on veterinary oncology” with the authors extensive background, other than attempting to cover the entire field of EP in oncology, with the following two reasons.

  1. The lack of focus

The authors attempted to cover almost the entire field of EP-based therapies on oncology, including irreversible electroporation (IRE), gene electrotransfer (GET), electrochemotherapy (ECT), calcium electroporation (Ca-EP), and therapeutic vesicles loading (i.e. liposomes, exosomes etc) (EP-TVL), which may take an entire book.

However, most of the areas have their corresponding insightful reviews already, and the reviewers can hardly find anything that hasn’t already been covered by other authors. For example:

  • Covering different techniques of electroporation (section 2 of this review)

Geboers, B., Scheffer, H. J., Graybill, P. M., Ruarus, A. H., Nieuwenhuizen, S., Puijk, R. S., ... & Meijerink, M. R. (2020). High-voltage electrical pulses in oncology: irreversible electroporation, electrochemotherapy, gene electrotransfer, electrofusion, and electroimmunotherapy. Radiology, 295(2), 254-272.

  • IRE (section 3.1)

Aycock, K. N., & Davalos, R. V. (2019). Irreversible electroporation: background, theory, and review of recent developments in clinical oncology. Bioelectricity, 1(4), 214-234.

  • GET (section 3.2)

Heller, L. C., & Heller, R. (2021). Gene electrotransfer. Electroporation in Veterinary Oncology Practice: Electrochemotherapy and Gene Electrotransfer for Immunotherapy, 219-234.

  • Calcium-EP (section 3.5)

Frandsen, S. K., Vissing, M., & Gehl, J. (2020). A comprehensive review of calcium electroporation—A novel cancer treatment modality. Cancers, 12(2), 290.

Romeo, S., Frandsen, S. K., Gehl, J., & Zeni, O. (2019, June). Calcium electroporation: an overview of an innovative cancer treatment approach. In 2019 PhotonIcs & Electromagnetics Research Symposium-Spring (PIERS-Spring) (pp. 2979-2984). IEEE.

  • EP-TVL (section 3.6)

Lennaárd, A. J., Mamand, D. R., Wiklander, R. J., El Andaloussi, S., & Wiklander, O. P. (2021). Optimised electroporation for loading of extracellular vesicles with doxorubicin. Pharmaceutics, 14(1), 38.

However, the veterinary applications of IRE, GET and Calcium-EP have not been extensively reviewed yet. Therefore, the reviewer would recommend shifting the focus a bit.

We thank for the constructive observations. Our intent was to provide a broader vision to readers that are not familiar with the potential of this methodology.

  1. Lack of original contribution

The following two reviews have covered the ECT from veterinary experience to human oncology exceedingly well. The reviewer has not able to identify too much new information on ECT in this reviewer that hasn’t been covered before, even though a few different examples were given.

Spugnini, E. P., Menditti, D., De Luca, A., & Baldi, A. (2023). Electrochemotherapy in translational medicine: from veterinary experience to human oncology. Critical Reviews™ in Eukaryotic Gene Expression, 33(1).

Spugnini, E. P., & Baldi, A. (2019). Electrochemotherapy in veterinary oncology: state-of-the-art and perspectives. Veterinary Clinics: Small Animal Practice, 49(5), 967-979.

We thank the reviewer, thanks to the observations of the other reviewers, the article has been implemented with additional information, making it more interesting for the audience.

Reviewer 3 Report

Comments and Suggestions for Authors

Dear Editor and Authors,

Thank you for asking me to review this work titled “Electroporation in translational medicine: from veterinary experience to human oncology” by Dr. Spugnini and colleagues.

In this review work the authors present the concept of electroporation an untested, undocumented, experimental procedure used in veterinary medicine and animals and propose its utilization in human subjects!! The authors try to present in a scientific, evidence based way what essentially remains an unproven treatment method based as much as allowable on supposition and magical thinking and less in science!! Naturally the work is unable to do this as there is no scientific basis for much of the assertions made.

As such I am extremely disinclined to propose acceptance of this work!! Kind regards to all.

Comments on the Quality of English Language

Dear Editor,

I did not want to write this out in the open but this is an unscientific, mambo jumbo area with minimal actual evidence. It is as much of value as snake oil!! I would not touch this work with a ten foot pole!!

Thank you,

Author Response

Reviewer 3

In this review work the authors present the concept of electroporation an untested, undocumented, experimental procedure used in veterinary medicine and animals and propose its utilization in human subjects!! The authors try to present in a scientific, evidence based way what essentially remains an unproven treatment method based as much as allowable on supposition and magical thinking and less in science!! Naturally the work is unable to do this as there is no scientific basis for much of the assertions made.

We have some problems to address this reviewer’s comments. Electroporation is widely used both in veterinary and human oncology, to the point that electrochemotherapy treatment is offered by National Health System and reimbursed by medical insurances….. Honestly we do not see where the “magic” aspect lays….

Reviewer 4 Report

Comments and Suggestions for Authors

The manuscript by Spugnini et al. raises the question of veterinary experience appropriate for translation to human oncology using electroporation techniques. The manuscript collects a number of interesting examples from veterinary practice and draws directions for clinical practice. I find this review interesting and timely. However, I would like to suggest that the authors work on the improvements that I hope will make their work even better.

- In my opinion, the introduction should include more information on the use of various approaches to the use of electricity in medicine. In particular, electrophoresis approaches that are also used for drug delivery should be mentioned. In the last paragraph, the authors should define what EP is in contrast to other approaches and that they will limit their work to EP.

-In my opinion, there is still room for more illustrations that summarize the most important points.

- I think it is better to avoid abbreviations within subchapter titles.

Author Response

Reviewer 4

- In my opinion, the introduction should include more information on the use of various approaches to the use of electricity in medicine. In particular, electrophoresis approaches that are also used for drug delivery should be mentioned. In the last paragraph, the authors should define what EP is in contrast to other approaches and that they will limit their work to EP.

We thank the reviewer for the suggestion. The article has been implemented with information about electrophoresis approaches for drug delivery.

-In my opinion, there is still room for more illustrations that summarize the most important points.

- I think it is better to avoid abbreviations within subchapter titles.

This point has been addressed

Round 2

Reviewer 1 Report

Comments and Suggestions for Authors

The Authors significantly improved the paper; there is a minor remark to be addressed: 

- drug distribution during ECT - there was mention of BLM concentration, but it would be beneficial to control this during the procedures, see examples:  doi: 10.1152/ajplung.00180.2022 and https://doi.org/10.3390/app13042027 

Author Response

This concept has been added as well as the reference

Reviewer 2 Report

Comments and Suggestions for Authors

There’s a substantial error in “Figure 1. A schematic depiction of the applications of EP in oncology.” IRE was not presented correctly.

As I pointed out previously, there is large overlapped (~1/2 of the entire article, mostly related to ECT) with the authors previous publications, most noticeably are:

·         Spugnini, E. P., Menditti, D., De Luca, A., & Baldi, A. (2023). Electrochemotherapy in translational medicine: from veterinary experience to human oncology. Critical Reviews™ in Eukaryotic Gene Expression, 33(1).

·         Spugnini, E. P., & Baldi, A. (2019). Electrochemotherapy in veterinary oncology: state-of-the-art and perspectives. Veterinary Clinics: Small Animal Practice, 49(5), 967-979.

The structure, layout and even the figures are identical. The contents are largely rewritten; the reviewer here does not imply self-plagiarism. However, the practice of recreating contents based on existing work without substantial new information is not encouraged during my training and in my institute. Even though some information is “updated”, for example with latest examples or clinical trials, this addition does not count as original but incremental. Most prestigious journals the reviewer took part in would strongly discourage this.

The reviewer understands that publishing in open-access journals will increase visibility, especially when subscription to these journals is only limited to selective institutes. For example, the reviewer had to resort to interlibrary loan to access the one published in “Critical Reviews”.

Therefore, the reviewer still hopes that the author can live up with a higher standard of scientific publishing, even though the work may already pass the quality bar of MDPI journals.

Author Response

REVIEWER 2

There’s a substantial error in “Figure 1. A schematic depiction of the applications of EP in oncology.” IRE was not presented correctly.

 The figure has been corrected

As I pointed out previously, there is large overlapped (~1/2 of the entire article, mostly related to ECT) with the authors previous publications, most noticeably are:

  • Spugnini, E. P., Menditti, D., De Luca, A., & Baldi, A. (2023). Electrochemotherapy in translational medicine: from veterinary experience to human oncology. Critical Reviews™ in Eukaryotic Gene Expression, 33(1).
  • Spugnini, E. P., & Baldi, A. (2019). Electrochemotherapy in veterinary oncology: state-of-the-art and perspectives. Veterinary Clinics: Small Animal Practice, 49(5), 967-979.

The structure, layout and even the figures are identical. The contents are largely rewritten; the reviewer here does not imply self-plagiarism. However, the practice of recreating contents based on existing work without substantial new information is not encouraged during my training and in my institute. Even though some information is “updated”, for example with latest examples or clinical trials, this addition does not count as original but incremental. Most prestigious journals the reviewer took part in would strongly discourage this.

THE article has been modified as requested. The figures are novel (different clinical cases and original cartoon)

The reviewer understands that publishing in open-access journals will increase visibility, especially when subscription to these journals is only limited to selective institutes. For example, the reviewer had to resort to interlibrary loan to access the one published in “Critical Reviews”.

Therefore, the reviewer still hopes that the author can live up with a higher standard of scientific publishing, even though the work may already pass the quality bar of MDPI journals.

We thank for the suggestion.